# IMOVE—An Intuitive Concept Mobility Systems for Perioperative Transfer and Induction of Anaesthesia for Special Needs Children

**DOI:** 10.3390/s20174901

**Published:** 2020-08-30

**Authors:** Hwan Ing Hee, Kiang Loong Ng, Manolo STA Cruz, Aloysius Tan, Haoyong Yu

**Affiliations:** 1KK Women’s and Children’s Hospital, Department of Paediatric Anaesthesia, Duke-NUS Medical School, The Anaesthesiology and Perioperative Sciences, Singapore 169857, Singapore; 2HOPE Technik Pte Ltd., Singapore 608614, Singapore; ngkiangloong@hopetechnik.com (K.L.N.); manny@hopetechnik.com (M.S.C.); aloysiustan@hopetechnik.com (A.T.); 3Department of Biomedical Engineering, National University Singapore, Singapore 119077, Singapore; bieyhy@nus.edu.sg

**Keywords:** mobility assistance, healthcare workplace safety, special needs children, autistic spectrum disorder, sensor and actuator

## Abstract

Children with autistic spectrum disorder (ASD) often exhibit uncontrollable disruptive behaviour during transfer to the operating room and operating table and at the induction of anaesthesia (sleep). This process often involves the physical restraining of children. These children are then lifted onto the operating table by healthcare staff after being anaesthetized. This predisposes children to fall risk and hospital staff to musculoskeletal injuries. We developed two concept mobility devices, IMOVE-I and -II, based on robotics systems comprising of restraint modules and multi-positional modality (sitting, supine, Trendelenburg). The aim was to intuitively secure children to facilitate the safe induction of sleep and ease of transfer onto operating tables upon sleep. IMOVE-I loads the child in standing position using a dual arm restraint module that is activated by trained healthcare staff. IMOVE-II loads the child in the sitting position by motivating the self-application of restraints. Opinions were obtained from 21 operating theatre healthcare staff with experience in the care of ASD children and parents with ASD children. The mean satisfaction rating of IMOVE-I was 5.62 (95% CI 5.00, 6.27) versus 8.10 (95% CI 7.64, 8.55) in IMOVE-II, *p* < 0.001. IMOVE-II is favoured over IMOVE-I in system operation and safety, ease of use and module functionality.

## 1. Introduction

Children with autism spectrum disorders (ASD) are known to have impairment in social communication skills and exhibit repetitive sensory–motor behaviours [1,2,3,4,5,6]. The presence of multiple sensory stimulation (light, smell, sound) and departure from familiar routines in the hospital environment are potential stressors for children with ASD. Many of these children become stressed and anxious in hospital environments, this can lead to maladaptive behaviour, such as aggression or the “fight or flight state”, that is beyond their control [1]. This disruptive behaviour often poses a challenge during routine hospital processes, such as moving from one ward to another, moving to the operating room, sitting or lying on the operating table and the induction of general anaesthesia (deep sleep state) [6,7,8]. Parents and healthcare staff and even children themselves are at risk of physical and psychological trauma when children resist, struggle to escape or fight.

The first line management of anxious and uncooperative special needs children includes the use of conservative interventions, such as cognitive behavioural modification (distraction, persuasion), preoperative psychosocial intervention (preparation with social stories), play therapy and parental presence [1,2,3,5]. When these fail, sedation medications are given to alleviate anxiety and facilitate cooperation. The physical restraint of children by parents and healthcare workers is often the last option when treatment and procedures are required [3,8,9]. However, this can further agitate an already stressed and frightened child, triggering or aggravating a combative behaviour.

When these children refuse to lie or sit on the operating table during induction of anaesthesia, they are manually lifted and carried by healthcare staff onto the operating table only after sleep is induced. When these children become older and heavier, physical restraint, human manual lifting and the carrying of these children become difficult; greater manpower may be required. This predisposes children to risk of fall and hospital staff to risk of back injury.

This is an emerging problem, with rising incidences of ASD worldwide. In the United States, the estimated prevalence of ASD school-going children above 8-years-old is 1 in 88 [1]. At the same time, staff injuries arising from managing “combat” patients and the manual lifting of ASD children has also gained attention as an area of concern. The high rate of injuries sustained by healthcare workers has been highlighted by the Occupational Safety and Health Administration (OSHA), United States [10]. More than 50% of injuries and illnesses reported among nurses and nursing support staff were related to musculoskeletal disorders [10,11,12]. The most common cause of injuries was lifting and straining [13,14,15]. In particular, the Association of periOperative Registered Nurses (AORN) Workplace Safety Task Force had identified transferring patients on and off operating tables and lifting and holding patients’ extremities [13,14] as high-risk perioperative tasks for musculoskeletal injury in perioperative nursing. Another area deserving attention is the use of physical restraints in this group of children. While physical restraint may bring about psychological and physical harm [3,8], it is often necessary to prevent significant harm to patients during the medical treatment and procedures. Hence, medical restraint is defined as one being used to promote medical and surgical healing and may be used in situations when a patient’s behaviour threatens the physical safety of the patient, staff, or others [16,17]. Commercially available restraints used in healthcare setting include belts (e.g., wheelchair belt), safety vest, posy vest, papoose board and limb straps. Complications from the use of these conventional mechanical restraints include fall and fracture arising from the patient struggling out of an ineffective restraint, poor circulation at sites of restraint, and in severe cases choking and asphyxia [18,19]. It is therefore important to explore a safe and effective mode of restraint that can improve both patient safety and patient experience.

To enhance patient safety and experience, staff protection and efficiency of intra-hospital transfer process, a specialized concept mobility system was designed and developed by our team, which comprises of clinicians and engineers. This endeavour took us from an experimental model, IMOVE-0 to two further prototypes—the IMOVE-I and IMOVE-II systems. To meet the needs of patient safety and patient experience in clinical situation, the mechanism of the safe restraint system evolved from one that is based on robotic automated restraint aided by pressure sensors in IMOVE-0 to machine assisted restraint in IMOVE-I and then self-assisted restraint in IMOVE-II. The principle of the IMOVE systems aim to intuitively engage the patient in a gentle and secure manner to aid perioperative movement. The goal is to achieve safe induction of anaesthesia and enable ease of transfer onto operating tables upon sleep with the use of an effective restraint system and an efficient multi-positional configuration. The unique human-machine design should allow rapid and easy patient loading and at the same time stabilize and secure the patient. The engineering challenge of this project lies in this interactive mobility system. We presented parts of the IMOVE-I prototype [20] in a conference paper. In this paper, we present the restraint system in the experimental IMOVE-0 model and the impetus in the design iteration to IMOVE-I and IMOVE-II, and compare the IMOVE-I and II systems.

## 2. Method

The proposed Concept Mobility System is a robotic system with sensing and actuation systems that adopts a human centric robotics approach [21,22] to achieve the following physical attributes and functions:

A. Required physical attributes.

(1) A child friendly exterior to appeal to children or to blend in with the operating room (OR).

(2) A small footprint for ease of storage and agility in an already congested hospital environment.

(3) A sitting/supine position as low as possible to reduce the impact of an accidental fall, thereby increasing patient safety.

B. Required Functionality.

(1) Ability to hold on to the child gently yet firmly to protect the child from falls and injuries.

(2) Ability to transition into various positions required for patient transfer and medical procedure (such as supine position for insertion of airway devices or Trendelenburg position for oropharyngeal suctioning).

(3) Ability for height adjustment to reach at least the minimum height of commercially available operating tables to facilitate transfer to operating tables.

(4) Ability to maneuver with ease in a crowded OR.

(5) Stability during transitions in all positions, device movement and during “combative” states of child.

Four main modules were designed in the system to achieve the above aims:

(1) Multi-positional main body module: This provides support for the patient in three basic functional positions—a resting engaging position, a supine bed position and a Trendelenburg position. The resting position is the initial position to engage or load the patient. This may be an “erect standing mode” or a “sitting chair mode”. The supine-bed position is the position for the parallel transfer of patient to the OT table; this position also allows for the maintenance of the patent upper airway (including application of airway maneuver such as the head-tilt/chin-lift or jaw-thrust and the insertion of airway devices). The Trendelenburg position allows airway protection maneuvers, such as suctioning of oropharyngeal content, when the patient is anaesthetised.

(2) Restraint module: This is the module that secures the patient safely in an efficient manner employing the principle of safe physical machine interaction [23].

(3) Mobility module: This is the chassis of the system that provides both stability and the mobility of the device. The mobility module is configured to transform in dimension (length and height) to facilitate the transition between various positions of the main body module. It also houses the electronics and power module. The weight distribution of the chassis is designed such that the system remains stable in all positions.

(4) Electronics and power module. This module includes the batteries, the controller electronics that power the movement of the multi-position bed module and motorize the configuration of the mobility system.

The following parameters were defined to scope the technical performance of the concept mobility system:Patient weight <50 kg;Patient height <1.6 m;Restraint module to take 50 kg separation force.

### 2.1. Description of the IMOVE-I and IMOVE-II Systems

The early design of IMOVE system (Experimental IMOVE-0 and Prototype IMOVE-I) was to mimic (a) the restraining and holding action of an uncooperative child by healthcare staff and parents during the induction of anaesthesia, and (b) the manual transfer of the child after sleep is induced. The restraint system here simulates a series of human actions that takes place in clinical situations. The experimental prototype, IMOVE-0 examined the possibility of robotic actuated automation restraint aided by pressure sensors. IMOVE-I, is the prototype iteration after the experimental model. Here, machine-assisted restraint is achieved that is largely controlled by human operation. The concept of IMOVE-I system was presented to network of parents with special needs children, allied healthcare staff (occupational therapist and physiotherapist) and operating theatre staff. Inputs obtained were considerations included in the cycle of design iteration and ideation development process that resulted in a newer IMOVE-II prototype.

The concept mobility systems (IMOVE-I and IMOVE-II) were developed with different methods of patient engagement and loading into the system (Figure 1).

(1) IMOVE-0 and IMOVE-I: The operation requires both parental cooperation and coordination by an experienced and trained healthcare provider in the operating theatre setting for the safe deployment of the restraint arm. The latter could include anaesthesia nurses, anaesthetists or operating theatre (OT) assistants. The design of both systems engages the patient by adapting to their typical combative posture in the standing position. In IMOVE-0, the healthcare provider activates the restraint module by demand and the degree of snugness is delivered by the robotic system and its pressure-sensing system. In IMOVE-I, the healthcare provider initiates the engagement of the restraint module and controls the magnitude of the securing force. Once restraint is in place, the machinery trigger to initiate the sitting position is activated by the operator.

(2) IMOVE-II. The child plays the active role in engagement with the system in the “resting chair” mode, this includes sitting in the “chair” and self-placement of the restraint module. The exterior look of IMOVE-II is designed to resemble a hybrid between a car seat, computer and massage chair, items that a child is familiar with and can relate to in an every-day situation. The amusement ride seatbelt and racing car seat design further adds an element of fun to alleviate anxiety in the hospital setting for the children.

### 2.2. Dimension

The dimension of IMOVE-I and IMOVE-II is illustrated in Figure 2. The height of the chair–bed configuration in IMOVE-I varies between 600 and 750 mm, while that for IMOVE-II varies between 450 and 750 mm.

The dimension of the IMOVE-I in erect resting position is similar to that of a typical radiation protection mobile X-Ray lead screen (1800 mm high and 750 mm), a common equipment found in the operating room. The dimension of the IMOVE-II in sitting resting position is similar to that of an office chair, an item that is commonly found in the operating room.

### 2.3. Multi-Positional Main Body Module

The multi-positional main body module of IMOVE-I is illustrated in Figure 3. IMOVE-I has 4 positional configurations (erect standing, sitting, supine, Trendelenburg position). The front and rear frame slides freely relative to each other with the use of drawer slides. The transformation of the configurations is brought about by 5 actuators. The conversion of the standing position of a child to a sequential sitting and supine position is achieved through a unique design of the machine-man interaction via the activation of an actuator. The triggering of the actuator results in its extension, which pushes the front frame forward. This forward motion gentle nudges the back of the child’s knee to bring the child into sitting position. By increasing the size of the base, the stability of the structure is further enhanced. Upon reaching the sitting position, four latching rods at the seat frame are secured in the latches. From the sitting position, the latches at the front frame are released electronically, upon which the same actuator retracts, causing the seat to recline and putting the patient in the supine position. The Trendelenburg position of up to 15 degrees is achieved by extending the length of a paired actuator by 70 mm in excess of another pair of actuators. Through the control pendant, the height of the bed module can be adjusted by controlling the extension of the paired actuators.

The Multi-positional main body module of IMOVE-II is illustrated in Figure 4. A commercial computer gaming seat was disassembled to provide seat and back support to form the chair and bed support of the IMOVE-II bed (see Figure 4b) in this project. IMOVE-II has 4 positional configurations (sitting, supine, lateral side-tilt, Trendelenburg position).

From the sitting to supine position, a dual electric actuator is used to extend and retract the rear casters to increase the stability of the system by achieving a 300 mm travel extension/retraction. This moves the back seat to recline into the supine position. The Trendelenburg position, to a maximum of 15 degrees head down, is achieved by extending the length of the same actuator that is used to recline the back rest. Through the control pendant, the height of the bed module can be adjusted by controlling the extension of the electro-hydraulic actuator using a scissor lift configuration. An added feature in the IMOVE-II is the ability of bi-directional side tilt mechanisms to facilitate the function of patient transfer parallel to the OT table. This feature is achieved by implementing a bi-directional electronic rotary latch and solenoid for the end limit stopper.

The transition speeds to and from various positions in the 2 systems are illustrated below in Table 1.

### 2.4. Restraint Module

The restraint module in IMOVE-0 is illustrated in Figure 5.

The possibility of automation of the restraint process with pressure sensing was evaluated in an experimental prototype. The design of the restraint module comprised of (a) a structural dual arm restraint system that was reinforced with (b) a pneumatic system that was made up of soft inflatable air pockets placed around the restraint arms. The set-up of the restraint system and the pneumatic system in the experimental prototype is illustrated in Figure 5. The idea of integrating the arm restraint with a cushioning pneumatic (air-bag) system was to reduce impact to the child’s torso during restraining and to facilitate even distribution of applied force. This can potentially increase patient safety and comfort and minimize psychological shock, while maintaining a secured and firm hold. During deployment, (a) the height of the dual arms can be adjusted and (b) the air pockets can be inflated in a controlled speed and pressure based on the pressure sensor information. The air-pockets are actuated by a portable pneumatic system powered by batteries in the electronics and power module. The compressor is powered to pressurize the accumulator to 3 bars (43.5 psi). When the pressure is achieved, the compressor is turned off and ball valve closed. During the activation of the air-bags, the solenoid valve opens and allows the air from the accumulator to pressurize the airbags to 0.1 bar (1.5 psi). In addition, an option of speed of restraint arm deployment was added to allow flexibility to customize to the need of the child in the clinical situation. In the experiment, the testing restraint force used in each restraint arm was 5 Newtons, air-bag inflation pressure of 0.1 bar (1.5 psi). The two ranges of adjustable speed of arm deployment were 1.7 to 2.5 s.

The restraint module in IMOVE-I and IMOVE-II systems is illustrated in Figure 6.

The restraint module in IMOVE-I system is designed with the aim to allow the operator to control the tightness, thereby ensuring the safety and effectiveness of the restraint system. The restraint module is a dual arm restraint module that is made up of 2 arms coupled to the restraint handle; a safety strap (seat belt) is harnessed to the 2 arms laterally and the handle at the back of the IMOVE-I system. This restraint system is manually deployed by pulling down on the handle with the operator standing behind child. The snugness of the strap over the patient can be tightened by pulling from the side of the handle at the rear, through a mechanism similar to that of a bag strap. This module is height adjustable such that the restraint arms can fit the height and position of the child to achieve secure restraint. This result in the descend of both arms over the patient in a motion affected by gears; this also brings the strap forward across the patient to harness the child. Upon securing the harness, the device is activated to bring the patient into sitting position.

The restraint module in the IMOVE-II system is designed with the aim of engaging children in self-motivated participation. The restraint module is composed of three options, a strap restraint (seat belt) across the hips, an “amusement park” overhead chest restraint and combination of strap belt integrated with the overhead chest restraint. This allows the child to self-apply the restraint module. The degree of snugness of the strap belt to securely restrain the child is adjustable either by the child or operator through a mechanism similar to that of a car seat strap belt.

### 2.5. Mobility Module

The IMOVE-I is fitted with omni-directional caster wheels to allow free rotation in the two horizontal axes. The rear wheels are caster wheels and can be lever locked. This configuration allows for maneuverability and braking. IMOVE-II is fitted with four lockable caster wheels so that it can similarly be pushed in any horizontal direction.

### 2.6. Electronic Module

Both systems are powered by 12 V 24 Ah AGM Lead Acid Batteries, which are sufficient to operate the systems for 11 h before charging.

The IMOVE-I system is controlled by three programmable motor controllers, Roboteq MDC2460. Each motor controller controls up to 2 actuators and 4 electronic latches. The man–machine interface is provided by a wired pendant to allow the operator to communicate with the controllers. The controller collects inputs from the user and controls the actuators. These inputs include the commands to move to bed position and height adjustments.

IMOVE-II system is controlled by a direct switch-motor with additional relay to protect the low power switch. There is no motor controller or micro controller, hence the system is simpler and low cost in comparison. A wired pendant allows the operator to communicate with the motors through a bulkhead connector and via switches that correspond to the intended actuator motion.

### 2.7. Survey

A prospective observational cross-sectional survey was conducted in a Specialist Children’s Hospital to obtain feedback on the design of the concept mobility systems, IMOVE-1 and IMOVE-II. The study proposal (CIRB 2016/2413) was reviewed and determined by the institutional review board that a formal review was not necessary for the survey feedback. Participants in the survey were parents of ASD children and healthcare professionals who worked in the operating theatre setting and had experience with the care of children with ASD. Exclusion criteria included lack of English proficiency. The participants were shown photographs of the IMOVE systems and videos of the operations of the 2 systems. Participants were asked to give written free response feedback with regard to the functionality of the modules and system operations, efficiency, safety, and ease of use. They were also asked to rate their satisfaction level of the 2 systems for perioperative transfer and the induction of anaesthesia in children with ASD. Opinions on advantages, disadvantages and need for additional support for the use of the system were also obtained. Fisher exact test was used for comparison of frequency and t test was used for the comparison of continuous outcomes between groups. A *p* value of <0.05 was taken as significant.

## 3. Results

The automated restraint system in the experimental model, IMOVE-0 was evaluated by a team of nine engineering staff playing roles of operators and users of the system. All agreed that the deployment and control of the restraint is at least easy (Likert scale of five in terms of operations and the control of arm-closure movement, arm-height adjustment, air bag and speed of deployment. Their responses as users are presented in Figure 7.

Twenty-one participants completed the survey comparing IMOVE-I and IMOVE-II. The mean age was 32.6 years (95% CI 29.7, 35.4). Participants were 10 operating theatre nurses (anaesthesia nurses and dental nurses), 7 doctors (anaesthesia, dental surgeon and general surgeon), 1 Operating theatre assistant and 3 parents with children diagnosed with ASD.

The results of participant’s opinions and ratings of potential operational, functionality, efficiency, ease of operation and safety of systems are presented in Table 2. The overall satisfaction of IMOVE-I was 5.62 (95% CI 5.00, 6.27), and IMOVE-II was 8.10 (95% CI 7.64, 8.55), *p* < 0.001. Additionally, three parents agreed to participate but did not complete the survey due to time constraints. All gave favourable verbal approval of the IMOVE-II system.

A summary of advantages and disadvantages of IMOVE-I and IMOVE-II opined by participants are presented in Table 3 and Table 4, respectively.

Additional support, such as the installation of more restraint straps (such as lower limb straps) and the attenuation of noise during the transition of positions were suggestions for both systems. Other suggestions for IMOVE-I included the use of a weighted blanket and manpower to hold the legs, especially for obese or small and hyperactive children, improving its exterior appeal/attractiveness, and the addition of cushions. Harnesses designed with graduated “snugness intensity” was also opined to lessen the “frightening” impact on children. Other suggestions for IMOVE-II included the addition of sensory feeding features, such as surface projection at the arm rests and feet rests to alleviate anxiety in ASD children and adjustable seat restraints to cater for different child sizes.

## 4. Discussion

While psycho-cognitive deficit and social impairment are well recognized hallmarks of ASD, abnormalities in motor function and development in children with ASD have not received as much attention. Many children with ASD have impaired motor development [1,4,24,25,26] and difficulty in planning and coordinating movement, often resulting in clumsiness and odd motor gait [3]. The safe transportation of a child with special heath in the hospital setting is an area of practice that is not widely known or investigated. Research exploring ways to improve transportation and the use of restraints during the induction of anaesthesia in children with ASD is also limited. We hope that the IMOVE systems would meet the challenges and needs of these children in the peri-operative hospital environment. This is especially relevant when children with ASD have an increased rate of hospital contact, compared to children without ASD [27], as they are also be more likely to require anaesthesia for medical procedures.

We previously described the mechanism of IMOVE-I [20]. IMOVE-I was designed to mimic the current mode of physical restraint of combative children in the hospital setting. This often involves parent or healthcare staff “hugging” or holding the child from the back, assisted by other healthcare staff at the side or front of the child. The deployment of IMOVE-I relies on the operation of the system by trained healthcare staff and requires the coordination of the deployment of the device with parents. The main critique of the design lies in the complexity involved in the preparation of the restraint strap over the dual restraint arms, the initiation of the restraint module and the accuracy and speed required to successfully secure the target. This sequential action requires a reasonably acquiescent target, sufficient distraction and quick appropriate positioning of the child by a team of parents and healthcare staff, as well as the coordination between the operator and the team of parents and healthcare staff. Thus, healthcare staff must be trained and competent in the deployment of the system; parental preparation and involvement is equally important. While conventional human manual holding of children is intuitive, the conversion of human interaction to a machine assisted interaction adds complexity and challenges. Many participants agreed that IMOVE-I may be effective in securing and moving uncooperative and combative children, thereby achieving the goal of facilitating a medical procedure. However, many participants also highlighted the safety concerns of the restraint. Another concern is the potential negative patient experience arising from involuntary restraint by the mechanical or machinery modality. This intimidating experience may result in psychological trauma which may negatively influence future surgical or hospital visits.

The mean satisfaction rating of IMOVE-I is significantly lower than IMOVE-II. Significantly more participants rated IMOVE-II more favourably compared to MOVE-I in system operation and safety, ease of use and module functionality. Unlike MOVE-I, IMOVE-II is designed with inputs from parents of ASD children who are involved in the care of their children. We engaged parent opinions in the design of IMOVE-II to gain valuable insight into the demands and challenges in the care of these special-needs children. Parents of ASD children are experts in the knowledge of their children’s needs, the triggers for agitation, as well as the solutions to alleviate a meltdown condition [1,3,9]. They are widely recognized to be the best resources for developing individualized strategies [28] for controlling children’s disruptive behaviours during a meltdown tantrum. IMOVE-II adopts an individual centred perspective that focuses on the child-device interaction and leverages on child’s self-autonomy and interest in engaging with the mobility device in the context of fun and play. This potentially promotes better patient experience and begets cooperation from these children in repeated hospital procedures. However, the key to the success of such a concept rests on the willingness of the child to approach the device. Besides the exterior appeal of the device, familiarity with the device is essential. Parents have suggested that the latter can be achieved with early introduction of I-MOVE to children to encourage interaction before surgery and the use of social stories about IMOVE to prepare the children at home. IMOVE-II may also be useful in facilitating ambulation of the child from arrival in hospital to the operating theatre suite.

We designed the IMOVE systems by understanding the stressors presented to ASD children in the peri-operative period. During this period, negative child behaviour [5,28] may arise from their anxiety or their restricted stereotypical behaviours. This may be related to waiting processes and routine procedures associated with the hospital visit. These children are further agitated by hypersensitivity to sensory stimuli, such as light, unfamiliar crowds, sounds, smells and tactile encounters, as well as movement. Being uncertain of their roles and their inability to understand instructions and inability to communicate their feelings, further compound the state of anxiety [1,2,3,5,6,28]. Despite these psycho-social challenges, many of these children are able to attend schools, sit through classes and take daily transportation on the road. They are familiar with the use of car seat belts and many are able to secure and apply seat belts by themselves. The IMOVE-II system leverages on the ability of ASD children to perform routine familiar tasks. The act of sitting on the “special chair” and putting on the restraint themselves, gives them a definitive role to play in the peri-operative period. This may help to minimize maladaptive behaviour by reducing their anxiety state through distraction and play. IMOVE-II may be better suited for ASD children who are motivated for self-care. While physical restraint should be avoided whenever possible [3,8,29], some amount of restraint may be necessary to promote the safety and best interest of these children, even in their daily lives [3,8,29]. The overhead “amusement park” restraint and seat belt in IMOVE-II together further provide protective stabilization in a non-threatening manner. While IMOVE-I may be intimidating, it may be the only way to help restraint severe ASD children who are combative if medical procedures cannot be postponed. The deployment of IMOVE-I would require caution to avoid harm to the children, parents or other healthcare staff in the proximity. IMOVE-I is thereby designed to work with the child in a passive role, while IMOVE-II strives to engage the child as an active participant in the human–device interaction.

In addition to the different restraint modules and modes of restraining, IMOVE-II is also more compact compared to IMOVE-I; it is therefore more advantageous in terms of mobility as well as storage in a limited operating room size that range from 400 to 800 square feet [30]. Its chair design also allows cushioning and greater comfort. IMOVE-II’s lower minimum height offers a greater margin of safety during any inadvertent accidental fall. It also has a lateral tilt that aids the parallel transfer for perioperative healthcare staff. IMOVE-II is favoured over IMOVE-I in system operation, safety, ease of use and functionality of its various modules.

There are potential ways in which sensing devices can be incorporated in the operation of IMOVE systems to improve patient safety and experience. These include the automatic sequential transformation from sitting to supine position in IMOVE-II that can be achieved via a weight sensor placed in the seat to detect the loading of the child. Pressure sensors can also be embedded in the restraint chest harness module in IMOVE-II or the restraining dual arms of IMOVE-I to allow graduated levels of tightness (low, medium, high). This has the potential to reduce physical harm from excessive compression of the restraining modules on the chest, minimize psychological trauma from a frightening patient experience due to the sensation of being “captured”. In the experimental stage, the possibility of automation of the restraint process with pressure sensor was evaluated. The use of 5 Newtons force and 0.1 bar air bag inflation were tested as an initial testing phase, which was not sufficient to establish a firm restraint. We did not proceed to increase testing with higher restraining forces or pressures in the air bag because it is difficult to ascertain the safest and optimal restraint force in the clinical situation during patient’s “fight or flight” stage. While the theoretical ideation of automation is novel and potentially useful, one should be cognisant that the automation of human–machine alone using a sensor system is challenging in uncertain clinical situations, such as those involving patients with Autistic Spectrum Disorder or patients with cognitive-behaviour conditions. These patients may exhibit unpredictable escape or combative behaviour that may disrupt the function of the restraint module and result in patient harm. These sensors system may be more beneficial as an additional feature to add comfort rather than as a primary mode of restrain. We have re-iterated further prototypes with restraint approaches that departs from the original idea of automation with sensing such as IMOVE-I based on machine assisted restraint and patient self-initiated restraints in IMOVE-II in order to achieve goals of patient safety and experience in clinical situations.

There are limitations in this study. Firstly, this is a prototype concept mobility device. Evaluation is not based on clinical use but rather on the clinical experience of management of children with ASD, as well as an understanding of the peri-operating challenge. We also did not evaluate opinions on the comfort of the systems, which are important aspects of patient experience. This could be explored in the future. The simulated use of IMOVE-I was performed with a volunteer acting as an ASD child and the successful deployment of the restraint module was carried out. We recognized that validation with a survey is not adequate to meet the standards for ergonomics and safety. The goal of this paper was to explore engineering solution and concept to load and secure a child who may be anxious and to determine the best approach. Further bench testing in a simulated environment will be performed with IMOVE-II system. Secondly, the dimensions of the prototypes were designed for older children (school-going age) and, below 50 kg, it is not intended for use in younger children or bigger children in the current form. Thirdly, engineering constraints in the prototypes include the sound generated by the motor, the speed of the transition of positions and the minimum height of the main body module. Noisy transition of positions and speed limitation during transitions were a result of the cost constraint of actuators used. Noise attenuation can be achieved with the use of higher performing actuators. The minimum sitting–supine position of IMOVE-II was limited by its small footprint and the compactness of the system; however, its minimum 450 mm height achieved was comparable to the range of minimum seat heights of office chairs (16–21 inches; 406 to 533 mm). Lastly, the response from our participants focused on patient safety rather than staff safety. We were not able to ascertain their opinions on the use of IMOVE systems on enhancing staff safety.

Aside from patient safety, staff welfare and wellbeing are important motivation for the IMOVE system. Physical stressors that occur with patient handling in the operating room (moving or lifting patients) have been reported to contribute to musculoskeletal disorder for perioperative healthcare staff. Musculoskeletal disorders are the most common reasons for long term absence from work for perioperative nurses [11,31]. The AORN provides guidance for implementing a “Safe patient handling and movement program”, and recommends forming an interdisciplinary team to select safe patient handling technologies and equipment, as well as to assess the unique needs of patients. The development of both IMOVE systems had, in broad principle, followed the guidance by taking in the opinions of major stakeholders, including perioperative healthcare staff, engineers and parents of ASD children. We focused on ergonomics by design for patient comfort, patient safety and ease of operator use. We integrated multi-purpose multi-position configuration to minimize manual transfer, handling and the restraining of patients. In future, IMOVE-II will be tested in a simulated environment to evaluate its potential clinical use.

## 5. Conclusions

A concept mobility system that enables transition into multiple positions and promotes self-autonomy in the human-device interaction is a potentially efficient approach to enhance patient safety during induction of sleep and patient transfer in the perioperative hospital setting for children with Autistic Spectrum Disorder.

## 6. Patents

HI Hee, Yu HY. SingHealth. 2019-07025. Publication of WO29194331A1. A personal transportation apparatus. https://patents.google.com/patent/WO2019143301A1. Application Number 10201800576V. eFile ref number: E201801220392T. Date: 22 January 2018.

## Figures and Tables

**Figure 1 sensors-20-04901-f001:**
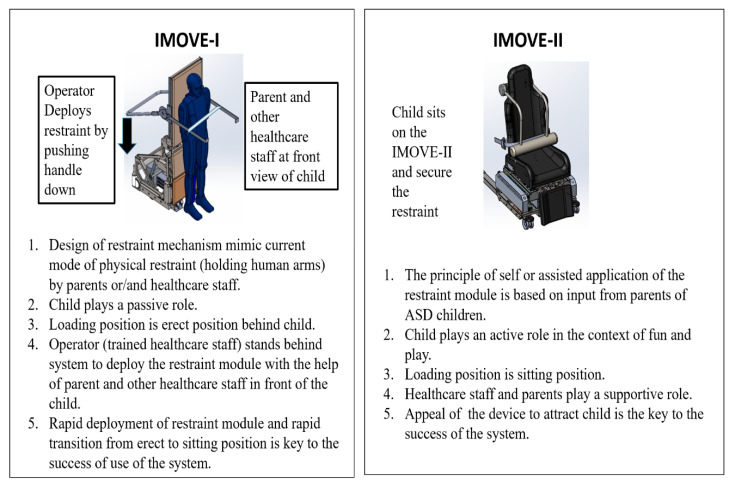
Methods of loading and engaging child in IMOVE systems.

**Figure 2 sensors-20-04901-f002:**
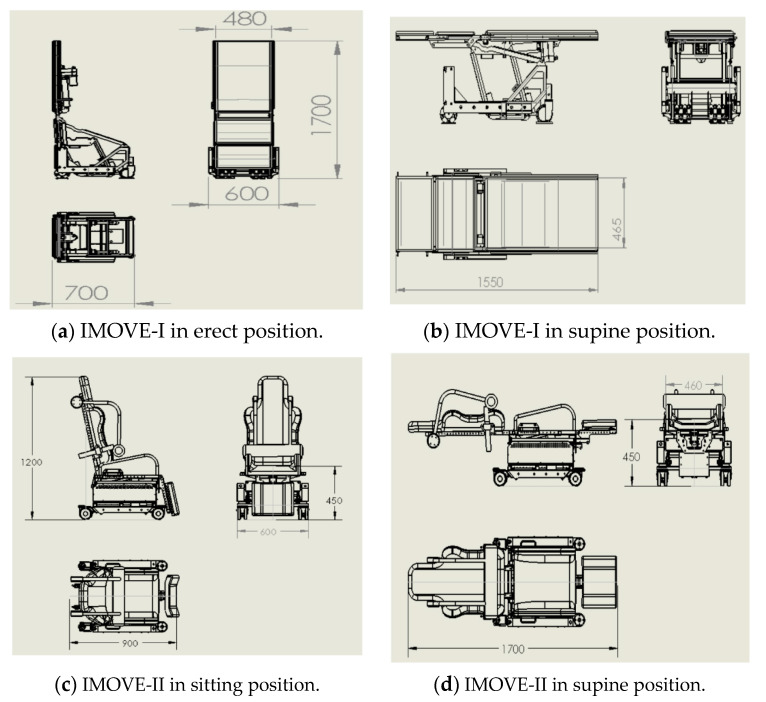
Dimension of IMOVE-I and IMOVE-II in mm. (**a**) IMOVE-I in the erect position. (**b**) IMOVE-I in the supine position. (**c**) IMOVE-II in the sitting position. (**d**) IMOVE-II in the supine position.

**Figure 3 sensors-20-04901-f003:**
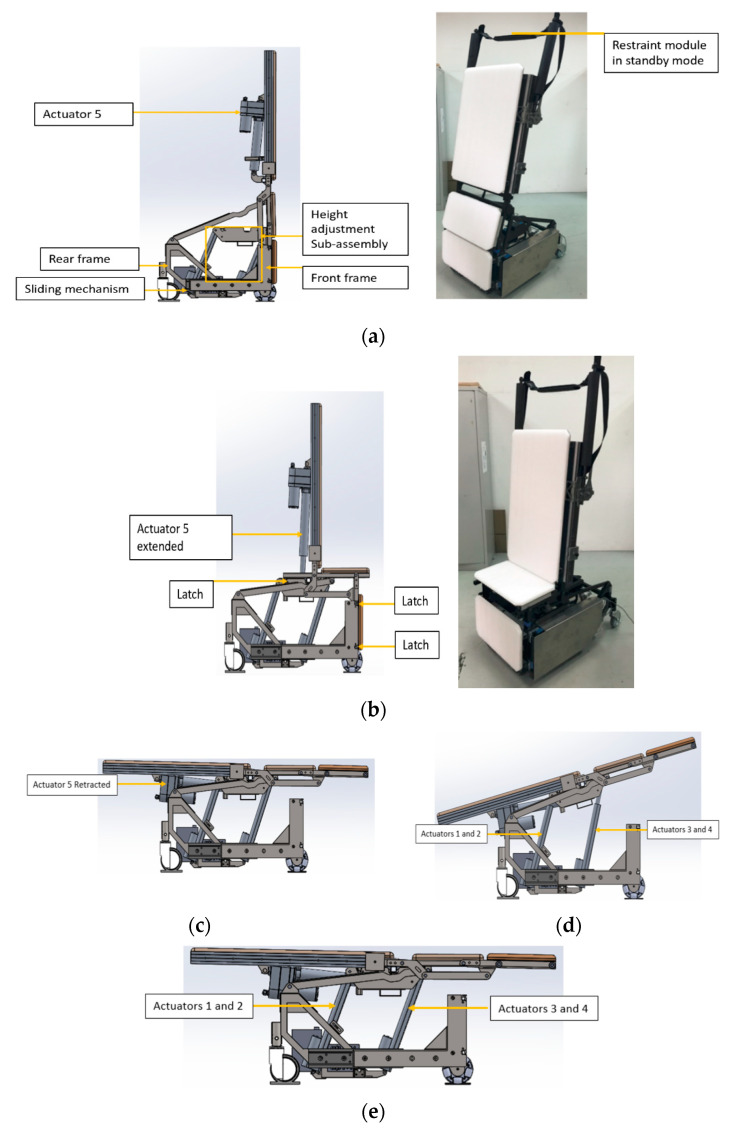
Multi-position of IMOVE-I. (**a**) Resting erect standing position, photo on the right. (**b**) Sitting position, photo on right. (**c**) Supine position. (**d**) Trendelenburg position. (**e**) Height adjustment.

**Figure 4 sensors-20-04901-f004:**
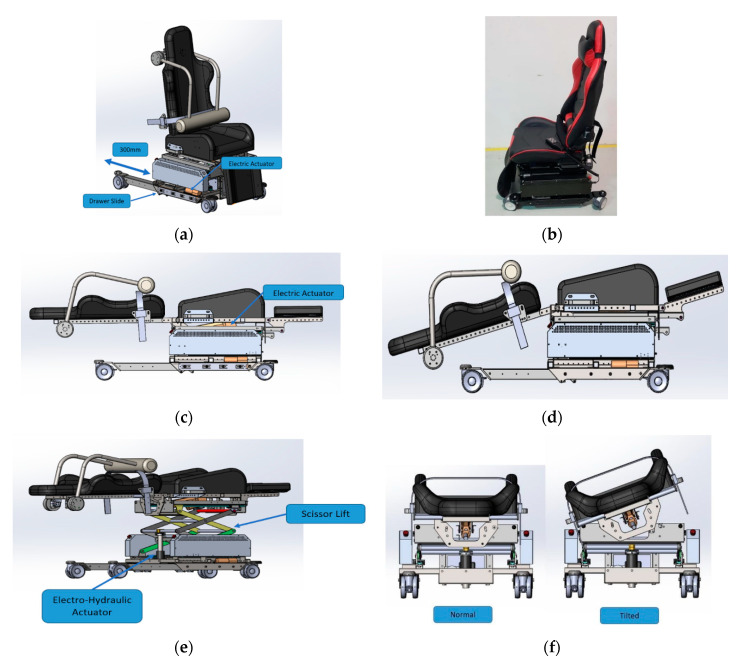
Multi-position of IMOVE-II. (**a**) Resting sitting position. (**b**) Photo on right without restraint module. (**c**) Supine position. (**d**) Trendelenburg position. (**e**) Height adjustment. (**f**) Lateral tilt position.

**Figure 5 sensors-20-04901-f005:**
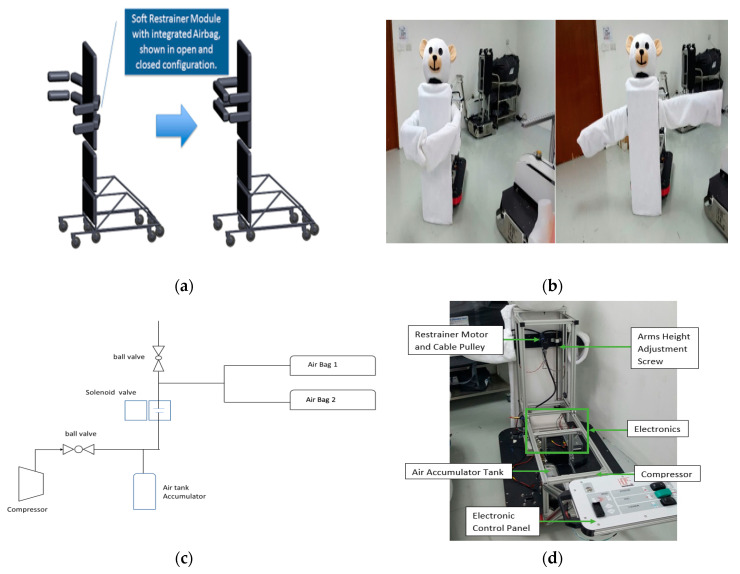
Restraint module of experimental model, IMOVE-0. (**a**) Restraint module comprising of dual restraint arm and arm air-bags. (**b**) Photo of the Arms Open and Close Position in the experimental model. (**c**) Schematic diagram of the pneumatic system in the dual air-bags. (**d**) Structure of the restraint motor and pneumatic system.

**Figure 6 sensors-20-04901-f006:**
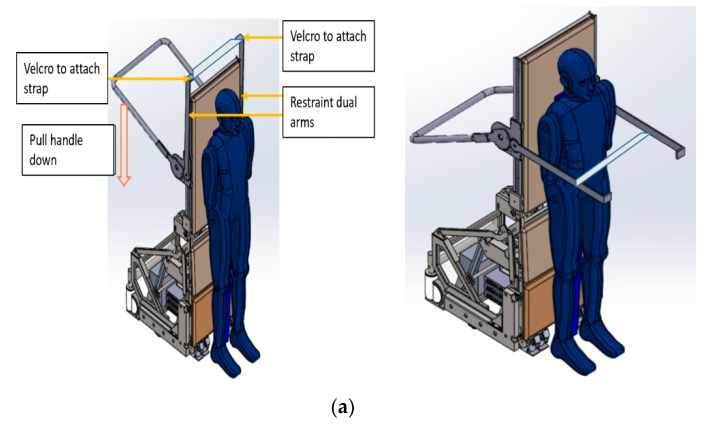
Restraint module of IMOVE systems. (**a**) Restraint arms in standby ready position and deployed position in IMOVE 1. (**b**) Restraint arms in standby ready position and deployed position in IMOVE-II.

**Figure 7 sensors-20-04901-f007:**
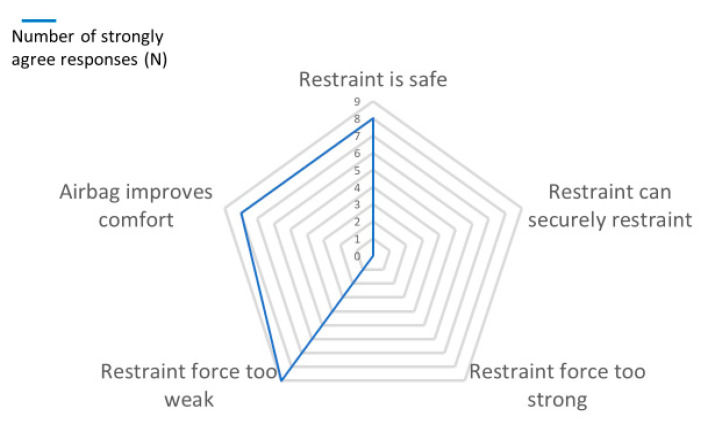
Evaluation of the restraint system by engineers.

**Table 1 sensors-20-04901-t001:** Transition time between positions. NA; not applicable.

No.	Original Position	Transformed Position	Time Taken (s)	
			IMOVE-I	IMOVE-II
1	Standing	Sitting	10	NA
2	Sitting	Supine	10	30
3	Supine	Trendelenburg 15 degrees	26	8
4	Supine (castor retracted)	Supine (castor extension)	NA	4

**Table 2 sensors-20-04901-t002:** Opinions on system operation, system efficiency, system safety, ease of use of system, functionality of modules and the requirement for additional support. Conditional: subject to conditions or requirements being met to be agreeable.

	IMOVE-I N (%)	IMOVE-II N (%)	*p* Value
Operationality of system			*p* < 0.001
Disagree	9 (42.9)	0 (0.0)	
Agree	7 (33.3)	17 (81.0)	
Conditional	5 (23.8)	3 (14.3)	
Not sure	0 (0.0)	1 (4.8)	
Functionality of modules			*p* < 0.001
Disagree	10 (47.6)	0 (0.0)	
Agree	7 (33.3)	19 (90.5)	
Conditional	3 (14.3)	2 (9.5)	
Not sure	1 (4.8)	0 (0.0)	
Efficiency in perioperative process			*p* = 0.002
Disagree	8 (38.1)	1 (4.8)	
Agree	6 (28.6)	17 (81.0)	
Conditional	5 (23.8)	1 (4.8)	
Not sure	2 (9.5)	2 (9.5)	
Ease of use			*p* < 0.001
Disagree	8 (38.1)	0 (0.0)	
Agree	7 (33.3)	20 (95.2)	
Conditional	4 (19.0)	0 (0.0)	
Not sure	2 (9.5)	1 (4.8)	
Safety of operations			*p* < 0.001
Disagree	12 (57.1)	1 (4.8)	
Agree	1 (4.8)	18 (85.7)	
Conditional	6 (28.6)	2 (9.5)	
Not sure	2 (9.5)	0 (0.0)	
No Additional support required			*p* = 0.002
Disagree	15 (71.4)	5 (23.8)	
Agree	3 (14.3)	13 (61.9)	
Maybe/not sure	3 (14.3)	3 (14.3)	

**Table 3 sensors-20-04901-t003:** Summary of advantages of IMOVE-I and IMOVE-II systems.

**Advantages of IMOVE-I**
*Form:* Looks like an OT table, easier to shift the child. *Loading Position*: Operator and device behind child, less intimidating for child. Loading does not require child to be familiar with the device. *Ease of use and control with* Remote control. *Child focus*: Allows more focus on child safety by minimizing manpower to restraint child and manual transfer while minimizing staff injury. Useful for uncooperative bigger kids.
**Advantages of IMOVE-II**
*Form*: Attractive, appealing, high tech, looks cool and comfortable, child friendly, familiar to child. The “arcade/roller coaster”, “cushioned race car seat”, “computer gaming chair” “padded chair” feature attracts child to sit in it. Children and parents would see it as a normal chair. Child may relate it to everyday normal item and maybe more cooperative. It also gives child a feel of going on an adventure. It is easier to attract child to sit in the “chair” using distraction activities, such as computer games or toys attached to the chair. It can also be used as a part of a game. Useful for children who can be coaxed with games, music, toys, bean cushions. Child may be given this transport system in operating theatre waiting area, during sedation and then transferred to OT seamlessly. Feasible if there are sessions before operation to get children to be familiar with the chair and the seats and parents are willing to take the time to bring child. Steadier and more stable than IMOVE-I. *Loading position*: Engaging child in the sitting position makes induction of anaesthesia easy. The sitting position is also useful to immediately allow placement of child in supine position after sedation. Sitting position is also safer even in a fall. *Ease of operation and control*: system is ergonomic and user friendly. (a) Easy to learn, operate, manipulate and use and more intuitive. Remote is simple and easy to understand. (b) Child is already sitting with the restraint secured. (c) Does not require many hands to operate the system. *Restraint*: The roller coaster seat restraint (overhead shoulder) gives good support to hold the child securely and efficiently. This gives extra protection over the chest in additional to over the hip car-seat belt which is similar to car seat belt. Child maybe familiar with putting on seat belt themselves. *Compact*: Smaller system which maybe more easily stored in the OT. Easy to maneuver. *Patient Experience*: Improve patient experience by decreasing unhappiness preoperatively. Allow children to be brought into OT with less force if children are well entertained with the system. *Patient:* Chair is safer for toddlers and mild ASD patients. Use can be extended to adult patients, such as dementia patients and other neuron-typical children. This will work well as a dental chair and non-clinical use in school and shopping centre. *Staff protection*: Save manpower during induction and manual carrying and minimise risk of staff injury.

**Table 4 sensors-20-04901-t004:** Summary of disadvantages of both IMOVE-I and IMOVE-II systems.

**Disadvantages of IMOVE-I**
*Design*: Unattractive, not child-friendly. Looks intimidating, unwelcoming, unsteady. *Loading position*: Requires care to correctly position device, patient and operator for safe use. Requires child to be sufficiently distracted and standing still in front of the device. Children’s co-operation may be difficult especially if they may be running or rolling on the floor. Loading position makes it difficult to maneuver and chase after an escaping running child. *Multi-position*: Sudden movement from standing to sitting movement is potentially frightening for the child. *Operations*: Operations of the restraint strap and remote are complicated. The restraint module requires time to prepare and necessitates operator’s familiarization with the module. Use of the system may be a challenge for the operator, training for competency is required. (a) Deployment of system requires operator to stand behind it to trigger the restraint mechanism; speed and accuracy is essential for safe deployment. The operator may not have a direct clear view of the child and may require communication with another healthcare worker in front of child to deploy the system. This may become difficult when child becomes aggressive or struggles. This is potentially dangerous if the restraint strap does not “catch” to secure child correctly or straps the child in incorrect position. During a struggle, the side arm of restraint module may hit the head of child or the child may hit against the structure of device. Risk is increased in obese autistic child and operation is functionally more difficult in obese children. (b) Further addition of lower body straps would make deployment more complicated (c) There is uncertainty about how much strength is adequate to tighten the shoulder strap. (d) May require a few people to operate the system. *Restraint*: Restraint module maybe insufficient in a combative child. More restraints, such as over leg/hip are required. May not be suitable over a range of sizes—e.g., taller versus smaller patient. May be risky if the belt lands on incorrect position, such as the upper torso or neck. *Size*: Bulky and space occupying in operating theatre (OT) for use and storage. May need to have several sizes to accommodate children of various sizes. *Child Experience*: Maybe traumatizing for the child to be suddenly restrained from behind. Restraint may cause child to feel threatened and struggle more. The process may be a frightening experience like a punishment, leading to psychological trauma. Child may be too frightened for a repeated visit and use.
**Disadvantages of IMOVE-II**
*Loading:* Child cooperation and participation to sit in the chair is required. Time may be taken for the kid to be willing to sit on the chair. Child may not be attracted to sit on it in the first place. *Form*: The shoulder strap may be in the way on the head end, thus requiring more space to maneuver. May need to be wider and solid for steadiness. May need a range of chair sizes or adjustable seat restraints to cater to range of height and size. May be risky for small autistic patients or obese big children. *Restraint*: Current design of the seat belt may not be good enough to hold patient down and keep stationary. Good to have additional safety belts for the legs.

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
