# Peer review of "IMOVE—An Intuitive Concept Mobility Systems for Perioperative Transfer and Induction of Anaesthesia for Special Needs Children"

_sensors, 2020, doi:10.3390/s20174901_

Round 1

Reviewer 1 Report

The paper is highly interesting, presenting a technical solution for a significant problem, such as patient safety, particularly for children with ASD.

The document is complete, clear, and well written.

I suppose that the authors have ethical approval for the study. It should be incorporated into the text.

The validation only with a survey can be the weakest point of the paper, due to there are different standards for ergonomy and safety that should be applied in the testing phase.

Some minor details should be improved, such as put a description inline 93 about 'OT space' or lines 417/418 in the references.

Author Response

  1. I suppose that the authors have ethical approval for the study. It should be incorporated into the text.

Response : Yes. We submitted the study to the institutional ethics board for IRB approval and was reviewed by the IRB as not required for further review and study to proceed. This is stated in line 276.  

  1. The validation only with a survey can be the weakest point of the paper, due to there are different standards for ergonomics and safety that should be applied in the testing phase.

Response : We thank the reviewer for the comment and we have included this in our limitation (line 414-419 in revised manuscript). As we mentioned in line 405, this is a prototype concept mobility device. Evaluation is not based on clinical use but rather, on clinical experience of management of children with ASD as well as understanding of the peri-operating challenge.

  1. Some minor details should be improved, such as put a description inline 93 about 'OT space' or lines 417/418 in the references.

Response : Yes. We have made the amendments in discussion line 398 (revised manuscript) and added in a reference (reference 30 in revised manuscript).    

Reviewer 2 Report

The paper addresses the problem of in-hospital safe transportation and restraint during anaesthesia in children with ASD by presenting the design concept of two mobility systems, IMOVE-I and IMOVE-II. Twenty-one participants among parents of ASD children and healthcare professionals were surveyed on aspects such as system operation, efficiency, safety, ease of use, functionality of modules and requirement for additional support.

The biggest problem of this paper is that no technical content of interest to the sensor community is presented.

The two concept mobility systems (IMOVE-I and IMOVE-II) are presented, detailing various aspects, such as main characteristics of the two systems (section 2.1), their dimensions (section 2.2), description and illustration of the multi-positional main body module (section 2.3), description and illustration of the restraint modules (section 2.4), description of the mobility module (section 2.5), some basic information about the electronic module (section 2.6). However, the only scientific contribution is the prospective observational cross-sectional survey (section 2.7, section 3) aimed to obtain feedback on the design of the concept mobility systems. But no scientific aspects of interest to the sensor community were addressed. Therefore, this reviewer disagrees with the publication of this paper in Sensors.

Author Response

Response :

We thank the reviewer for the comment. The goal of this paper is to explore engineering solutions to load and secure an anxious combative child. In one of our approach, IMOVE-I system, we use a mechanical trigger from the robotic system (in the standing position) to nudge the lower limbs of child into sitting position and at the same time activate the restraint module to secure the child. Once the child sits on the machine, a change into supine bed position is activated by a human trigger via remote.  

There are 2 potential ways in which sensing devices can be incorporated in the operation of IMOVE-I to improve patient safety and patient experience.

  • The sequential transformation from sitting to supine position can be achieved automatically with a weight sensor placed in the seat.
  • Pressure sensor can also be embedded in the restraint module to allow graduated levels of tightness (low, medium, high) to reduce potential harm from excessive compression of the restraining dual arm straps on the chest and minimize psychological trauma from a frightening patient experience.   

The sensor design was not implemented here as this was an early prototype in which the engineering focus was on achieving a safe restraint combined with multi-position configuration utility with a small foot print.

We have included this in discussion line 402 to 408 in the revised manuscript.

Thank you.

Reviewer 3 Report

Hello, thank you for your work. 

I think that the participants in the questionnaire are no in significant numbers.

I believe that this system can also be used for many people with physical difficulties or a type of disability other than autism.

Why autism? Can you better explain why autism, please?

Kind regards.

Author Response

  1. Comment : I think that the participants in the questionnaire are no in significant numbers.

Response : We thank the reviewer’s comment and we agree. The aim of the study is to present and compare the mechanism used in the 2 mobility systems. We did not carry out a large cohort study but instead choose to focus on a small group survey for a qualitative type of response as both devices were prototype. A larger cohort study comprising of parents, children followed focusing on IMOVE-2.                 

  1. Comment :I believe that this system can also be used for many people with physical difficulties or a type of disability other than autism. Why autism? Can you better explain why autism, please?

Response :  We thank the reviewer’s comment and we agree. In this project we focus on the needs of children with autism. Unlike neurotypical children, children with autism face difficulty in coping and adapting to unfamiliar and new environment, in addition to difficulty in communicating their thoughts, fears and needs, many of them also have movement disorder. Perioperative period is challenging experience for them, the operating theatre is a new and a potentially threatening environment. This is summarized in paragraph 1 in the discussion and in introduction.    

Firstly, children with autism spectrum disorders (ASD) are known to have impairment in social communication skills and repetitive sensory–motor behaviours. Presence of multiple sensory stimulation (light, smell, sound) and departure from familiar routine in the hospital environment therefore presents challenges to children with ASD. Many of these children become stressed and anxious in hospital environment such as moving into the operating room and at induction of anaestheisa, this can result in maladaptive behaviour such as aggression, “fight or flight state” that is beyond their control. This disruptive behaviour often poses a challenge during routine hospital processes such as moving from one ward to another, moving to operating room, sit or lie on the operating table, induction and recovery from general anaesthesia (deep sleep state). Parents and healthcare staff and even children themselves are at risk of physical and psychological trauma when children resist, struggle to escape or fight      

While psycho-cognitive deficit and social impairment are well recognized as the hallmark of ASD, abnormalities in motor function and development in children with ASD have not received as much attention. Many children with ASD have impaired motor development, difficulty in planning and coordinating movement resulting in clumsiness and odd motor gait. The safe transportation of child with special heath in the hospital setting is an area of practice that is not widely known or investigated. Research exploring ways to improve transportation and use of restraint during induction of anaesthesia in children with ASD is also limited. We hope that the IMOVE systems would meet the challenges and needs of these children in the peri-operative hospital environment. This is especially relevant when Children with ASD have an increased rate of hospital contact, compared to children without ASD, they will also more likely to require anaesthesia for medical procedures.

Reviewer 4 Report

Manipulating persons with Autism spectrum disorders (ASD) is difficult. The difficulty may result in critical outcomes when manipulation occurs before anesthesia. The authors present here an adjustable and versatile set up that could prevent the difficulties induced when patients with atypical behavior when they are immobilised. The MS is focused on ASD, but the setup could be used with other patients. The illustrations are clear and help to understand the description and the results are not disputable. The MS has not an outstanding scientific interest, but its clinical usefulness is excellent.

Two modifications might be done.

Is it possible to insert here the reactions of the patients or their evaluation of the comfort in the set up?

What are the statistical tests? I suppose that the khi square was used. Is the effect size large? (the authors should compute the effect size and the confidence intervals). The new statistics must be discussed in the final part of the MS.

Author Response

  • Is it possible to insert here the reactions of the patients or their evaluation of the comfort in the set up?

Response : We thank the reviewer for the  pointer. We did not evaluate the specific reactions of the patients or comfort of the set-up in this survey and we have included this important point in the limitations (line 411-412 in revised manuscript). We recognized this is an important aspect especially for patient experience that would be evaluated in future.        

  • What are the statistical tests? I suppose that the khi square was used. Is the effect size large? (the authors should compute the effect size and the confidence intervals). The new statistics must be discussed in the final part of the MS.

Response : We thank the reviewer for the comments. Fisher exact test and t test were used for frequency and for continuous data (line 286 to 287 in revised manuscript). The statistics are presented in the result section as well as in the table 2 and discussion (line 355-356 revised manuscript). The reasons for the difference is explained in the discussion (lines 382 to 408).         

Round 2

Reviewer 2 Report

The previous comment of this reviewer was that no technical content of interest to the sensor community is presented in this paper.

In response, the authors have simply indicated (in Section 4. Discussion, lines 402-408 of the revised manuscript) two potential ways in which sensing devices can be incorporated. However, they did not perform any experimentation on an implementation of this idea.

Thus, the previous conclusion of this reviewer is still the same: no scientific aspects of interest to the sensor community are addressed in this paper.

This reviewer, therefore, believes that the manuscript is not worth publishing in Sensors.

Author Response

We thank the reviewer for the comment to improve the manuscript.

In the first experimental stage, the possibility of automation of the restraint process with pressure sensoring was evaluated in an experimental prototype. The design of the restraint module comprised of (a) a structural dual arm restraint system that is reinforced with (b) pneumatic system that is made up of soft inflatable air pockets placed around the restraint arms. The set-up of the restraint system and the pneumatic system in the experimental prototype is illustrated in Figure 6. The idea of integrating the arm restraint with a cushioning pneumatic (air-bag) system is to reduce impact to the child’s torso during restraining and to help evenly distribute the applied force. This can potentially increase patient safety and comfort, minimize psychological shock, while maintaining a secured and firm hold. During deployment (a) the height of the dual arms can be adjusted and (b) the air pockets can be inflated in a controlled speed and pressure based on the pressure sensor information. The air-pockets are actuated by a portable pneumatic system powered by batteries in the electronics and power module. See Figure 6C. The compressor is powered to pressurize the accumulator to 3 bars (43.5psi). When the pressure is achieved, the compressor is turned off and ball valve closed. During activation of the air-bags, the solenoid valve opens and allows the air from the accumulator to pressurize the airbags to 0.1 bar (1.5psi). In addition, an option of speed of restraint arm deployment was added to allow flexibility to customize to the need of the child in the clinical situation. In the experiment, the testing restraint force used in each restraint arm was 5 Newtons, air-bag inflation pressure of 0.1 bar (1.5 psi). The two range of adjustable speed of arm deployment was 1.7 to 2.5 seconds. Such an automation system was evaluated by a team of 9 engineering staff playing roles of operators and users of the system. All agreed that deployment and control of the restraint is at least easy (likert scale of 5) in terms of operations and control of arms-closure movement, arms height adjustment, air-bag and speed of deployment. Their response as users are presented in Figure 6D. The use of 5 Newtons force and 0.1 bar air bag inflation were not sufficient to establish a firm restraint, in fact when a child exhibits ‘fight or flight’, it is difficult to ascertain the best restraint force needed to secure safely without causing harm.       

                 6A                                 6B    

                  6C                                  6D

Figure 6. Restraint module of experimental model. 6A Restraint module comprising of dual restraint arm and arm air-bags. 6B. Schematic diagram of the pneumatic system in the dual air-bags. 6C. Structure of the restraint  motor, pneumatic system. 6D. Evaluation of the restraint system by engineers.     

While the theoretical ideation of automation is novel and potentially useful, one should be cognisant that automation of human-machine alone using sensor system is challenging in uncertain clinical situation such as one involving patients with Autistic Spectrum Disorder or patients with cognitive-behaviour conditions. These patients may exhibit unpredictable escapee or combative behaviour that may disrupt the function of the restraint module. These sensors system can be added features to add comfort. We have re-iterated prototypes with 2 different approaches, (1) machine assisted restraint in IMOVE-I. Here, added sensing devices for pneumatic can be incorporated in the operation of IMOVE-I to improve patient comfort and patient experience as demonstrated in early experimental prototype. (2) Patient initiated restraint in IMOVE-II to increase patient safety.     

This is added in discussion lines 409-448

Round 3

Reviewer 2 Report

In response to the previous reviewer's comments, the authors provided some material that may be of interest to the sensor community, i.e., an automated restraint system controlled by pressure sensing.

Although the added material involving pressure sensing can be considered somewhat innovative, at least, under the applicative point of view; this content still results excessively narrow with respect to the remaining of the manuscript which focuses on design presenting and surveying the two mobility device concepts, IMOVE I and II.

Thus, in order to balance contents, the new content regarding the automated restraint system controlled by pressure sensing should be further extended and spread among the various sections of the manuscript: 1. Introduction (providing motivations and comparison with state of the art involving the automated restraint system controlled by pressure sensing), 2. Method (moving there the figure 6 and related discussion, and describing the experimental setup involving the automated restraint system controlled by pressure sensing), 3. Results (presentation of experimental results concerning the automated restraint system controlled by pressure sensing), 4. Discussion (of results regarding the automated restraint system controlled by pressure sensing).

Author Response

We thank the reviewer for improving the manuscript

We have thus balanced the content by inserting the added material (with regards to an automated restraint system controlled by pressure sensing) in various section as recommended such as :

  1. Introduction (providing motivations and comparison with state of the art involving the automated restraint system controlled by pressure sensing),
  2. Method (moving there the figure 6 and related discussion, and describing the experimental setup involving the automated restraint system controlled by pressure sensing),
  3. Results (presentation of experimental results concerning the automated restraint system controlled by pressure sensing),
  4. Discussion (of results regarding the automated restraint system controlled by pressure sensing).
